# Edit-Based Refinement for Parallel Masked Diffusion Language Models

**Houxing Ren** [1]  **Mingjie Zhan** [1 2]  **Zimu Lu** [1]  **Ke Wang** [1]  **Yunqiao Yang** [1]
**Haotian Hou** [1]  **Junting Pan** [1]  **Hongsheng Li** [1 3 4]

## Abstract

Masked diffusion language models enable parallel token generation and offer improved decoding efficiency over autoregressive models. However, their performance degrades significantly when generating multiple tokens simultaneously, due to a mismatch between token-level training objectives and joint sequence consistency. In this paper, we propose ME-DLM, an edit-based refinement framework that augments diffusion generation with lightweight post-editing steps. After producing an initial complete response, the model refines it through minimal edit operations, including replacement, deletion, and insertion, conditioned on the full sequence. Training supervision is derived from edit distance, providing a deterministic signal under a fixed canonicalization scheme for learning minimal corrections. This approach encourages sequence-level consistency through globally conditioned edits while preserving the efficiency benefits of parallel diffusion decoding. Extensive experiments demonstrate that ME-DLM improves the quality and robustness of multi-token parallel generation. In particular, when built upon LLaDA, our method achieves consistent gains of 11.6 points on HumanEval and 33.6 points on GSM8K while using one-eighth of the total diffusion steps. Code is available at `https://github.com/renhouxing/ME-DLM`.

## 1 Introduction

Large language models (LLMs) (Zhang et al., 2026; Achiam et al., 2023; Islam & Moushi, 2025; Dubey et al., 2024; Liu et al., 2024; Anthropic, 2025; Yang et al., 2025a; Zeng et al., 2025; Comanici et al., 2025) have evolved from basic text generators into general-purpose systems that exhibit strong linguistic competence and broad task adaptability, performing well on a variety of natural language tasks such as translation (Zhu et al., 2024; Patil & Gudivada, 2024) and question answering (Matarazzo & Torlone, 2025). Recent models, including OpenAI o1 (Jaech et al., 2024) and DeepSeek-R1 (Guo et al., 2025), further strengthen this trajectory by improving reasoning and problem-solving abilities, allowing LLMs to address complex challenges such as competitive programming code generation (Jain et al., 2024) and Olympiad-style mathematics (He et al., 2024). Collectively, these advances position LLMs as increasingly general-purpose language and reasoning engines with broad applicability across real-world and scientific domains.

In parallel with advances in autoregressive LLMs, an alternative paradigm for language generation has emerged based on diffusion processes, commonly referred to as diffusion language models (DLMs) (Li et al., 2022; He et al., 2023; Chen et al., 2023b). Instead of generating text token by token in a strictly left-to-right manner, diffusion-based models operate by progressively denoising masked or corrupted token sequences, enabling inherently parallel prediction across positions. This property makes DLMs particularly attractive for fast decoding and scalable generation, as multiple tokens can be predicted simultaneously. Recent studies have demonstrated that mask-based diffusion approaches can achieve competitive performance with autoregressive models while offering substantial improvements in decoding efficiency and flexibility (Nie et al., 2025; Ye et al., 2025).

However, despite their promise, current diffusion language models often struggle when extending from single-token to multi-token generation. Empirically, when the model is required to predict multiple tokens simultaneously, such as generating 4 or 8 tokens in parallel, generation quality often degrades sharply. This degradation persists even in settings where the underlying training data contains valid and diverse target sequences, indicating a potential mismatch between token-level training objectives and coherent multi-token prediction. Understanding the origin of this failure mode and how to design diffusion-based language models that maintain correctness and consistency under multi-token parallel generation remains insufficiently understood.

---

[1]CUHK MMLab [2]SenseTime Research [3]Shenzhen Loop Area Institute [4]CPII under InnoHK. Correspondence to: Mingjie Zhan <zhanmingjie@sensetime.com>, Hongsheng Li <hsli@ee.cuhk.edu.hk>.

*Proceedings of the 43rd International Conference on Machine Learning*, Seoul, South Korea. PMLR 306, 2026. Copyright 2026 by the author(s).

In this paper, we find that this degradation is closely related to a discrepancy between marginal token prediction and joint sequence validity. Most masked diffusion language models are optimized with token-level cross-entropy objectives, which encourage accurate approximation of each token's marginal distribution conditioned on the input and diffusion state. While such token-level objectives only model marginal distributions, this limitation is largely masked in autoregressive decoding or single-token diffusion settings, where the joint sequence probability is implicitly factorized into a product of conditional distributions across steps. In these regimes, dependencies between tokens are captured through sequential conditioning, helping capture inter-token dependencies in practice despite token-wise training. In particular, when multiple tokens are selected simultaneously based solely on their individual marginal probabilities, an implicit conditional independence assumption is introduced across positions. Consequently, token combinations that are locally probable yet globally inconsistent can arise, even though all training sequences remain valid. This mismatch between marginal optimality during training and joint consistency during multi-token parallel generation offers an intuitive explanation for the degradation we observe.

To address the inconsistency arising from parallel multi-token generation, we propose a lightweight refinement strategy, ME-DLM, which augments masked diffusion with additional editing diffusion steps. After all tokens are unmasked and an initial complete response is produced, the model performs a subsequent edit-based refinement conditioned on the entire generated sequence. At this stage, the sentence is already largely well-formed, and the role of the edit step is to correct residual inconsistencies by applying a small number of localized operations. Concretely, the refinement process operates through a standard set of edit actions, *i.e.,* replacement, deletion, and insertion, allowing the model to transform the initial generation into a coherent and valid target sequence. To train the model to perform such refinements, we construct supervision signals by pairing noisy or imperfect generated sequences with their corresponding ground-truth responses. The minimal sequence of edit operations required to convert a generated sentence into the target is derived using edit distance, which provides a deterministic supervision signal under a fixed canonicalization scheme by enforcing the smallest possible modification set. This formulation enables the model to explicitly learn how to correct globally inconsistent outputs using minimal local edits, without requiring changes to the underlying diffusion generation process.

In summary, this work makes the following contributions:

- We observe that masked diffusion language models can suffer from degraded quality under parallel multi-token generation, and provide an intuitive explanation based on the mismatch between marginal token prediction and joint sequence coherence in such settings.

- We propose an edit-based refinement paradigm that augments diffusion generation with lightweight yet effective post-editing steps and an edit-distance–based training formulation that provides a deterministic supervision signal under a fixed canonicalization scheme, enabling learning of minimal, interpretable refinement actions.

- Extensive experiments demonstrate that the proposed approach consistently improves the quality and robustness of multi-token parallel generation. In particular, when built upon LLaDA, our method achieves consistent gains of 11.6 points on HumanEval and 33.6 points on GSM8K while using only one-eighth of the total diffusion steps.

## 2 Preliminaries

In this section, we introduce the fundamental components of the masked diffusion language model, followed by an analysis of the limitations of parallel multi-token generation.

### 2.1 Masked Diffusion Language Models

We consider language modeling as learning a generative distribution $p_\theta(x)$ over token sequences $x = (x_1, \ldots, x_L)$ of length $L$. Autoregressive large language models typically rely on an autoregressive factorization,

$$p_\theta(x) = \prod_{i=1}^{L} p_\theta(x_i \mid x_{<i}), \qquad (1)$$

which imposes a left-to-right generation order and models the joint distribution through sequential conditional predictions.

Masked diffusion language models (MDLMs) instead define $p_\theta(x)$ implicitly through a discrete diffusion process over token sequences. Given a clean sequence $x_0$, a continuous noise level $t \in [0, 1]$ is sampled, and a corrupted sequence $x_t$ is constructed by independently replacing each token in $x_0$ with a special [MASK] token with probability $t$. This defines a simple Bernoulli corruption process commonly adopted in discrete diffusion models for text. When $t = 1$, all tokens are masked; when $t = 0$, the sequence remains unchanged.

The reverse process is parameterized by a neural network $p_\theta(\cdot \mid x_t)$ that predicts the original tokens at masked positions conditioned on the partially observed sequence $x_t$. Training proceeds by minimizing a denoising cross-entropy objective over masked tokens:

$$\mathcal{L}(\theta) = -\mathbf{E}_{t,x_0,x_t} \left[ \frac{1}{tL} \sum_{i=1}^{L} \mathbf{1}[x_{t,i} = \mathbf{M}] \log p_\theta(x_{0,i} \mid x_t) \right],$$
$$(2)$$

where $\mathbf{1}[\cdot]$ denotes the indicator function and M denotes the mask token. The normalization factor $1/(tL)$ accounts for the expected number of masked positions. This objective can be interpreted as a variational denoising surrogate for maximum likelihood learning under the discrete diffusion framework, and has been shown to enable effective generative modeling of text.

**Inference.** Text generation in masked diffusion language models is performed by simulating the reverse denoising process starting from a fully masked sequence. Given a target length $L$, generation initializes from

$$x_1 = ([\texttt{MASK}], \ldots, [\texttt{MASK}]), \qquad (3)$$

and follows a predefined decreasing schedule $\{t_K > \cdots > t_0 = 0\}$. At each step $k$, the model predicts token distributions for all masked positions in $x_{t_k}$,

$$p_\theta(x_{0,i} \mid x_{t_k}), \text{ for } x_{t_k,i} = [\texttt{MASK}]. \qquad (4)$$

A subset of masked positions is then selected and filled according to these distributions, while the remaining positions stay masked, yielding $x_{t_{k-1}}$. In practice, the subset can be chosen either randomly or based on model uncertainty (*e.g.,* token entropy), following common heuristics in prior work. This iterative partial unmasking procedure continues until all tokens are revealed, producing the final generated sequence $x_0$. Unlike autoregressive decoding, this process does not impose a fixed generation order, allowing parallel prediction across positions.

## 2.2 Limitations of Parallel Multi-token Generation

Although masked diffusion language models enable parallel prediction across token positions, the combination of token-level denoising objectives and parallel decoding does not guarantee coherent sequence-level generation when multiple tokens are produced simultaneously. This limitation becomes particularly pronounced in multi-token parallel decoding settings.

The standard training objective optimizes a token-wise denoising loss, encouraging the model to approximate the marginal conditional distribution $p(x_{0,i} \mid x_t)$ at each position $i$. Crucially, this objective does not explicitly model or constrain the joint distribution $p(x_{0,1}, \ldots, x_{0,L} \mid x_t)$ over multiple positions. During parallel decoding, multiple tokens are generated simultaneously, often by independently selecting a token for each position based on its marginal distribution. For example, a common strategy is to decode a subset of positions $\mathcal{S}$ in parallel via

$$\hat{x}_{0,i} = \arg\max p_\theta(x_{0,i} \mid x_t), \quad i \in \mathcal{S}, \qquad (5)$$

which implicitly assumes conditional independence among

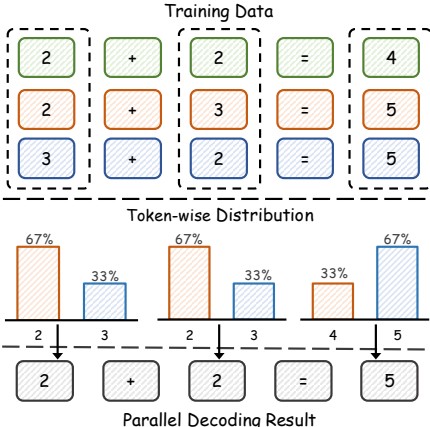

*Figure 1.* Illustration of failure in parallel multi-token generation.

the selected tokens. As a result, the induced joint distribution is approximated in a factorized manner:

$$p_\theta(x_{0,\mathcal{S}} \mid x_t) \approx \prod_{i \in \mathcal{S}} p_\theta(x_{0,i} \mid x_t). \qquad (6)$$

This factorized approximation can lead to token combinations that are individually likely under their marginal distributions but jointly inconsistent at the sequence level. Such inconsistencies may arise even when all training sequences are valid and diverse, as the training objective does not explicitly penalize invalid joint configurations that emerge from simultaneously selecting multiple tokens under a factorized decoding scheme. This mismatch between marginal optimality during training and joint consistency during parallel decoding offers an intuitive explanation consistent with our empirical observations.

Importantly, this issue is mitigated when tokens are sequentially generated or iteratively refined with conditioning on previously generated outputs, as conditional dependencies between tokens are progressively enforced. In contrast, under parallel generation with standard token-level objectives, the lack of explicit joint modeling constitutes a key limitation that becomes particularly pronounced under aggressive parallel decoding.

**Illustrative example.** Figure 1 illustrates the limitation of parallel multi-token generation through a concrete natural language example. Suppose the training dataset contains the following three valid sentences:

$$x^{(1)} = \text{``2 + 2 = 4''},$$
$$x^{(2)} = \text{``2 + 3 = 5''},$$
$$x^{(3)} = \text{``3 + 2 = 5''}.$$

All sequences are syntactically valid and satisfy the corresponding question–answer relations, while differing at

specific token positions. When trained with a token-level denoising objective, an MDLM learns marginal conditional distributions independently for each token position.

As a consequence, under parallel multi-token decoding, the model may generate a sequence such as

$$\text{``2 + 2 = 5''},$$

which combines high-probability tokens drawn from different valid training examples, yet violates the underlying sequence-level consistency constraint between the operands and the result.

This example highlights how parallel decoding based solely on marginal token predictions can produce outputs that are locally plausible but globally inconsistent. In contrast, sequential decoding or iterative refinement enforces conditional dependencies across tokens, thereby preventing such invalid combinations.

## 3 Methodology

Building on this observation in Section 2, we adopt a two-stage view of the denoising process. We first perform multi-token unmasking to obtain a coarse sequence that roughly matches the target at the semantic level, while potentially violating sequence-level consistency. Starting from this approximate sequence, we then reinterpret subsequent diffusion steps as a sequence refinement process based on editing operations. Under this formulation, token prediction serves to establish a reasonable initialization, whereas edit-based denoising is responsible for correcting structural inconsistencies that arise from parallel multi-token prediction. This separation allows the model to retain the efficiency of parallel unmasking while addressing its inherent limitations through targeted sequence refinement.

### 3.1 Edit-based Diffusion

We propose *edit-based diffusion* as an iterative refinement framework that improves a discrete sequence through parallel edit operations. Edit-based diffusion starts from an initial sequence and repeatedly applies local edits to progressively reduce errors over successive refinement steps.

Formally, let $x^{(t)} \in \mathcal{V}^{\leq L_{\max}}$ denote the sequence at iteration $t$. At each iteration, the model predicts a set of token-wise edit actions $\mathcal{E}^{(t)}$. These edits are applied deterministically through an edit application operator $A$, yielding $x^{(t+1)} = A\left(x^{(t)}, \mathcal{E}^{(t)}\right)$. A special *empty edit* action $\varnothing$ corresponds to predicting no changes to the current sequence. When the empty edit is predicted, the sequence remains unchanged, and the diffusion process terminates. In practice, we additionally impose a maximum number of refinement iterations to ensure termination, even if the model does not predict an empty edit.

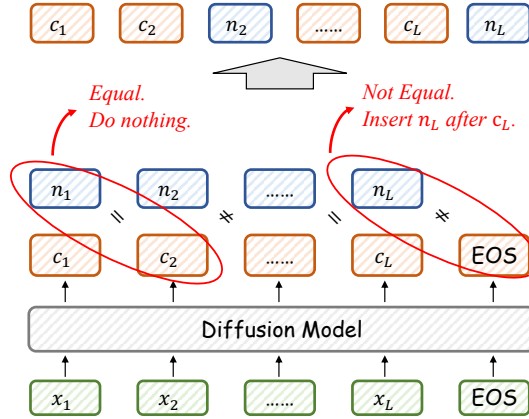

*Figure 2.* Illustration of edit-based diffusion refinement. The model operates on a complete sequence and predicts localized edit actions in parallel, enabling replacement, deletion, and insertion.

**Discussion.** Edit-based diffusion differs from standard diffusion processes in that it operates directly on complete sequences and applies discrete, localized edit actions. Its objective is not to approximate a target distribution asymptotically, but to progressively correct residual inconsistencies introduced by parallel generation.

Refinement actions are trained to be minimal and conservative, with supervision derived from the shortest edit scripts. This biases the model toward necessary local corrections rather than large-scale rewrites. As structural errors are resolved, the model increasingly predicts the empty edit action, and the refinement process naturally stabilizes once the sequence becomes self-consistent.

Although edit predictions are factorized at the token level, they are conditioned on the entire sequence and applied through a deterministic operator that couples decisions across positions. As a result, effective dependencies emerge at the sequence level through the shared edit application, rather than through the prediction distribution itself.

### 3.2 Model

To parameterize the refinement transition distribution $p_\theta(x^{(t+1)}|x^{(t)})$ in a manner that preserves parallelism, we model sequence refinement through token-wise edit predictions conditioned on the current sequence state. Formally, for each token position $i$, the model predicts a pair of variables $(c_i, n_i)$, where $c_i$ denotes the refined token at position $i$, and $n_i$ denotes a candidate token to be inserted immediately after position $i$. The variable $c_i$ takes values in the vocabulary augmented with a special deletion symbol [DEL], while $n_i$ takes values in the vocabulary. These token-wise predictions collectively define a stochastic edit operator that maps the current sequence $x^{(t)}$ to a refined sequence $x^{(t+1)}$.

Under this formulation, the transition distribution factorizes across token positions conditioned on $x^{(t)}$,

$$p_\theta(x^{(t+1)} \mid x^{(t)}) \equiv \prod_{i=1}^{L_t} p_\theta(c_i, n_i \mid x^{(t)}), \qquad (7)$$

while the deterministic application of the predicted edits induces structured dependencies in the resulting sequence. This factorization enables fully parallel prediction during refinement, while the induced edit semantics allow the model to express nontrivial sequence-level transformations across diffusion steps. As shown in Figure 2, given the token-wise edit predictions $\{(c_i, n_i)\}_{i=1}^{L_t}$ at diffusion step $t$, the refined sequence $x^{(t+1)}$ is constructed through a deterministic application of the predicted edits. The process proceeds in a single pass over the current sequence $x^{(t)}$ from left to right. For each token position $i$, the value of $c_i$ determines the fate of the current token $x_i^{(t)}$: if $c_i = [\texttt{DEL}]$, the token is removed; otherwise, $c_i$ replaces $x_i^{(t)}$ in the refined sequence. Subsequently, if $n_i$ is not equal to $c_{i+1}$, the predicted token $n_i$ is inserted immediately after position $i$. We further clarify how this edit application handles two boundary cases:

- **Repeated-token insertion.** Inserting an additional token $a$ between two adjacent tokens in $a\,a\,b$ yields $a\,a\,a\,b$, which can be equivalently represented by inserting $a$ between $a\,b$ under our canonical edit convention.

- **Boundary insertion.** Insertion before the first generated token is handled by treating the boundary between the prompt and the generated sequence as a valid editable position. This allows the model to insert tokens at the beginning of the generated response without introducing a separate boundary-specific operation.

### 3.3 Training and Inference

**Training.** Our objective is to train the edit-based refinement model to parameterize the diffusion transition kernel $p_\theta(x^{(t+1)} \mid x^{(t)})$. Given a ground-truth target sequence $x^\star$, we first obtain an initial noisy sequence $x^{(n)}$ via random masking and $n$ parallel multi-token unmasking diffusion steps. Starting from $x^{(n)}$, we perform $m$ steps of edit-based diffusion using the current model parameters, yielding an intermediate sequence $x^{(m)}$. This sequence serves as the noisy input state for training at diffusion depth $m$, and reflects the types of structural inconsistencies encountered during inference.

To construct supervision for edit prediction, we compute a minimal edit script that transforms the intermediate sequence $x^{(m)}$ into the target sequence $x^\star$. Specifically, we derive a sequence of replacement, deletion, and insertion operations corresponding to the edit distance between $x^{(m)}$ and $x^\star$. This edit script is then deterministically mapped to

---

**Algorithm 1** Training

1: **Input:** Token Sequence $x^\star = (x_1^\star, \dots, x_L^\star)$
2: Sample noise level $t \in (0, 1]$
3: Randomly mask tokens in $x^\star$ according to $t \to x^{(0)}$
4: Sample unmask diffusion steps $n$
5: Initialize $x \leftarrow x^{(0)}$
6: **for** $k = 1$ to $n$ **do**
7:      Randomly select a subset of masked positions
8:      Predict and unmask selected tokens to update $x$
9: **end for**
10: Sample edit diffusion steps $m$
11: **for** $k = 1$ to $m$ **do**
12:      Predict edit actions $\{(c_i, n_i)\}$ conditioned on $x$
13:      Update sequence $x \leftarrow \mathcal{A}(x, \{(c_i, n_i)\})$
14: **end for**
15: Compute minimal edit script between $x^{(m)}$ and $x^\star$
16: Map edit script to token-wise supervision $\{(c_i^\star, n_i^\star)\}$
17: Compute loss $\mathcal{L}(\theta; x^{(m)}, \{(c_i^\star, n_i^\star)\})$

---

token-wise supervision signals $(c_i^{(m)}, n_i^{(m)})$, which define the desired refinement actions at each token position. When multiple tokens need to be inserted at the same location, we supervise only the first insertion in the current refinement step and defer the remaining insertions to subsequent refinement steps. The model is trained to predict these edit actions conditioned on $x^{(m)}$, thereby learning to reduce sequence-level inconsistency through iterative correction. Although multiple shortest edit scripts may exist, we fix a canonical mapping to ensure consistent supervision during training. Empirically, we find the model insensitive to alternative equivalent scripts.

To stabilize training and mitigate compounding errors at early stages, we employ a curriculum over the diffusion depth $m$. At the beginning of training, we set $m = 0$, such that the model learns to correct inconsistencies directly from the initial unmasking output. As training progresses, the maximum diffusion depth is gradually increased, allowing the model to learn refinement behaviors at increasingly noisy and structurally diverse intermediate states.

**Inference.** The inference procedure closely mirrors the training process, differing only in the absence of supervision and parameter updates. Given an input prompt, we first obtain an initial noisy sequence via the same unmasking process used during training, followed by a fixed number of unmask diffusion steps to produce a coarse sequence. Starting from this initialization, we apply edit-based diffusion for up to a fixed maximum number of refinement steps, repeatedly sampling edit actions and applying the deterministic edit operator. Unlike training, where the diffusion depth is governed by a curriculum schedule and supervision is derived from edit distance to a reference sequence, inference

does not rely on reference targets and instead performs iterative refinement until either an empty edit action is predicted or the maximum refinement horizon is reached.

## 4 Experiments

In this section, we present extensive experiments to demonstrate the effectiveness of the proposed method and analyze its performance. Due to the limited space, more detailed experiments are presented in Appendix C.

### 4.1 Experimental Setup

**Training Dataset.** Our training procedure is built upon LLaDA-8B-Base[1] and consists of three sequential stages. Throughout all stages, we use a single model with shared parameters and perform full-parameter fine-tuning. The masked diffusion and edit diffusion processes differ only in their input states and decoding rules. In Stage 1, we train LLaDA-8B-Base for one epoch on Nemotron-Pretraining-SFT-v1[2]. Besides predicting the current token, the model is also trained to predict the subsequent token at each position, which provides the basis for token-wise edit modeling. This yields ME-DLM Stage 1. In Stage 2, we further fine-tune the model for one epoch on AM-DeepSeek-R1-0528-Distilled[3] using the standard masked diffusion objective only. This stage teaches the model to perform parallel multi-token unmasking and yields ME-DLM Stage 2. In Stage 3, we continue training on the same dataset with the proposed unmask-and-edit diffusion objective. This stage interleaves standard masked diffusion training with edit-based refinement training, enabling the model to preserve its parallel unmasking ability while learning to correct structural inconsistencies in complete draft sequences. The resulting model is denoted as ME-DLM Stage 3.

**Test Dataset.** We evaluate our method on four math and code tasks. For math tasks, we evaluate our method on GSM8K (Cobbe et al., 2021) and MATH-500 (Hendrycks et al., 2021). For code tasks, we evaluate our method on HumanEval (Chen, 2021) and MBPP (Austin et al., 2021). Considering the insufficiency of test cases in these benchmarks, we also use HumanEval+ and MBPP+ (Liu et al., 2023) for evaluation, which contain 80×/35× more tests.

**Implementation Details.** All experiments are conducted using the LLaDA architecture. To train the models efficiently, we employ DeepSpeed (Rajbhandari et al., 2020)

---

[1] https://huggingface.co/GSAI-ML/LLaDA-8B-Base

[2] https://huggingface.co/datasets/nvidia/Nemotron-Pretraining-SFT-v1

[3] https://huggingface.co/datasets/a-m-team/AM-DeepSeek-R1-0528-Distilled

*Table 1.* Hyperparameters of training.

| Parameters | Stage 1 | Stage 2 | Stage 3 |
|---|---|---|---|
| Learning Rate | 5e-5 | 5e-5 | 1e-5 |
| Batch Size | 2,048 | 128 | 128 |
| Sequence Length | 2,048 | 2,048 | 2,048 |
| Warmup Steps | 100 | 100 | 100 |
| Weight Decay | 0.01 | 0.0 | 0.0 |

and Flash Attention (Dao, 2023). For Stage 1 and Stage 2, we follow standard masked diffusion training. Given a target sequence, we sample a noise level $t \in (0, 1]$, randomly mask tokens according to $t$, and train the model to recover the masked tokens with a token-level objective. Stage 3 further introduces edit-based refinement training. To construct refinement supervision, we simulate the inference process: starting from a partially masked sequence, the model performs parallel multi-token unmasking without supervision, where the number of unmasked tokens per step is randomly chosen from $\{2, 4, 8, 16\}$. This process continues until no mask tokens remain, producing a noisy complete draft generated by the model itself. We then compute edit targets between this draft and the ground-truth sequence, and train the model to correct the draft through localized replacement, deletion, and insertion operations. This exposes the model to its own generation errors and better aligns training with the inference-time distribution. Other training hyperparameters are reported in Table 1. Stage 1 uses a larger batch size to support stable pretraining, while Stages 2 and Stage 3 adopt smaller batch sizes for supervised fine-tuning and edit-based diffusion training, respectively. All training is conducted on 64 NVIDIA H800 GPUs; Stage 1 takes approximately 150 hours, Stage 2 approximately 3 hours, and Stage 3 approximately 60 hours.

**Inference Details.** During inference, generation proceeds in two phases: mask diffusion first produces a complete coarse draft, and edit diffusion then refines it. We use a fixed step allocation strategy that prioritizes mask diffusion while reserving a small number of steps for edit-based refinement. By default, one quarter of the total diffusion steps is allocated to edit diffusion, with the number of edit steps capped at 32. Concretely, this gives mask/edit step allocations of 48/16, 96/32, 224/32, and 480/32 for total budgets of 64, 128, 256, and 512 steps, respectively. Unless otherwise specified, we use a maximum generation length of 512 tokens during inference.

### 4.2 Main Results

We include two representative prior approaches that focus on soft or relaxed masking for parallel generation, namely Soft Mask (Hersche et al., 2025) and EvoToken (Zhong et al.,

*Table 2.* Main results on reasoning and code generation benchmarks. Results are reported as Pass@1 for code tasks and accuracy for math tasks. Results of Soft Mask and EvoToken are copied from their papers. The best result under each setting is highlighted in bold. The budget is defined such that the product of the budget and the number of predicted tokens per step is kept constant, corresponding to a fixed total number of diffusion steps.

| Budget | Model | Evaluation Dataset | | | | | | Average |
|--------|-------|-----------|-----------|------|------|------|----------|---------|
| | | HumanEval | HumanEval+ | MBPP | MBPP+ | GSM8K | MATH-500 | |
| 1/1 | Soft Mask | 57.8 | 50.0 | 56.4 | - | 84.0 | 41.4 | - |
| | EvoToken | - | - | - | - | 84.5 | 41.0 | - |
| | LLaDA-Instruct | 44.5 | 37.8 | 49.5 | 40.7 | 71.9 | 27.6 | 45.3 |
| | ME-DLM Stage-2 | 56.1 | 50.0 | 54.8 | 46.8 | 80.9 | 45.8 | 55.7 |
| | ME-DLM Stage-3 | **57.9** | **53.0** | **61.6** | **52.9** | **84.8** | **50.0** | **60.0** |
| | Gain | +1.8 | +3.0 | +6.8 | +6.1 | +3.9 | +4.2 | +4.3 |
| 1/2 | Soft Mask | 38.3 | 33.8 | 48.4 | - | 79.4 | 38.8 | - |
| | EvoToken | - | - | - | - | 81.8 | 37.4 | - |
| | LLaDA-Instruct | 43.3 | 38.4 | 41.5 | 34.7 | 71.0 | 26.2 | 42.5 |
| | ME-DLM Stage-2 | 45.7 | 42.7 | 52.4 | 44.2 | 76.3 | 43.0 | 50.7 |
| | ME-DLM Stage-3 | **51.2** | **46.3** | **54.8** | **47.6** | **82.7** | **50.0** | **55.4** |
| | Gain | +5.5 | +3.6 | +2.4 | +3.4 | +6.4 | +7.0 | +4.7 |
| 1/4 | Soft Mask | 24.8 | 23.0 | 32.3 | - | 62.3 | 19.8 | - |
| | EvoToken | - | - | - | - | 72.3 | 31.2 | - |
| | LLaDA-Instruct | 29.3 | 25.6 | 26.7 | 23.0 | 66.0 | 23.4 | 32.3 |
| | ME-DLM Stage-2 | 36.6 | 32.9 | 34.7 | 31.0 | 58.9 | 32.0 | 37.7 |
| | ME-DLM Stage-3 | **42.1** | **39.0** | **45.0** | **37.6** | **76.6** | **38.2** | **46.4** |
| | Gain | +5.5 | +6.1 | +10.3 | +6.6 | +17.7 | +6.2 | +8.7 |
| 1/8 | LLaDA-Instruct | 12.2 | 9.8 | 17.5 | 15.3 | 50.3 | 20.2 | 20.9 |
| | ME-DLM Stage-2 | 13.4 | 13.4 | 22.0 | 18.8 | 30.2 | 17.8 | 19.3 |
| | ME-DLM Stage-3 | **25.0** | **22.6** | **26.7** | **22.8** | **63.8** | **34.4** | **32.6** |
| | Gain | +11.6 | +9.2 | +4.7 | +4.0 | +33.6 | +16.6 | +13.3 |

2026), as additional baselines. Some related methods, such as LRD (Zhu et al., 2025), dynamically adjust or terminate the number of inference steps by monitoring convergence during generation. As these approaches do not provide explicit control over the total number of diffusion steps and instead rely on adaptive stopping criteria, their inference budgets are not directly comparable to our fixed-budget setting in practice.

Table 2 presents the main experimental results comparing our method with these approaches as well as LLaDA-based baselines under different generation budgets across reasoning and code generation benchmarks. (1) Across most evaluation settings, the Stage-2 model consistently outperforms the LLaDA-Instruct baseline, indicating that supervised fine-tuning with diffusion-style unmasking provides a strong and effective initialization for parallel generation. We therefore adopt Stage-2 as the primary reference point for evaluating the effectiveness of our proposed method in subsequent comparisons. (2) Building upon the same Stage-2 initialization, our method consistently achieves higher performance across all budgets and datasets considered. This uniform improvement clearly demonstrates the effectiveness of edit-based refinement in enhancing multi-token parallel generation beyond standard unmask diffusion. Importantly, the observed gains span both code generation and mathematical reasoning tasks, suggesting that the proposed approach generalizes well across different forms of structured sequence generation. (3) Compared with prior parallel decoding baselines such as Soft Mask and EvoToken, our method achieves competitive or superior performance under carefully matched generation budgets. On GSM8K, at a budget of 1/2, our method shows a performance gap of 3.3 points over Soft Mask and 0.9 points over EvoToken. As the budget decreases to 1/4, these gaps further increase to 14.3 points and 4.3 points, respectively. This consistent widening of the performance gap across both baselines indicates that explicitly modeling sequence-level refinement through edit diffusion becomes increasingly beneficial as parallel decoding grows more aggressive. (4) Moreover, the benefits of our method become increasingly pronounced as the generation budget

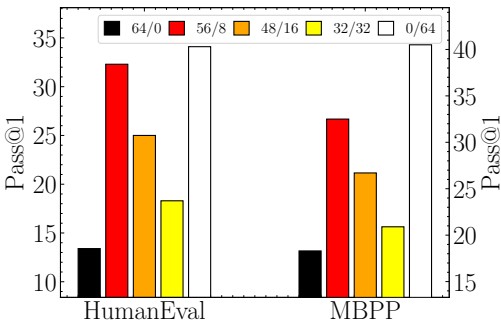

*(a) Code tasks.*

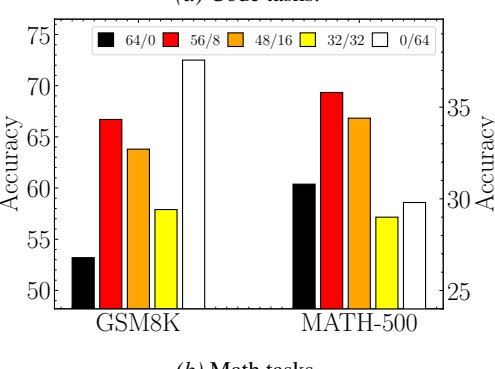

*(b) Math tasks.*

*Figure 3.* Effect of different allocations between mask diffusion and edit diffusion under a fixed generation budget of $1/8$ (64 total steps). Each legend entry $m/e$ indicates the number of mask diffusion steps and edit diffusion steps, respectively.

decreases. While the average improvement over Stage-2 remains modest at a full budget (1/1), it steadily increases as fewer diffusion steps are used, rising from an average gain of 4.3 at budget 1/1 to 13.3 at budget 1/8. This clear trend indicates that edit-based refinement effectively mitigates the degradation caused by aggressive parallel decoding, making it particularly well-suited for low-step, high-parallelism generation regimes.

### 4.3 Detailed Analysis

**Effect of Step Allocation.** To better understand how mask diffusion and edit diffusion contribute to generation quality under aggressive parallel decoding, we analyze different allocations of diffusion steps while keeping the total generation budget fixed. Specifically, we consider a budget of $1/8$, corresponding to 64 total diffusion steps, and vary the number of steps assigned to mask diffusion and edit diffusion, denoted as $m/e$ with $m + e = 64$. This setting allows us to isolate the effect of edit-based refinement when the overall number of refinement steps is severely constrained.

As shown in Figure 3, using only mask diffusion leads to a clear performance drop under such an aggressive decoding budget, indicating that parallel unmasking alone struggles to produce sequence-level consistent outputs with very few

*Table 3.* Analysis of the effective number of edit diffusion steps. We report the average number of edit steps until the predicted edit action becomes empty, indicating convergence. Results are shown under different generation budgets. Here, "HE" denotes "HumanEval" and "MATH" denotes "MATH-500".

| Budget | Total | HE | MBPP | GSM8K | MATH |
|---|---|---|---|---|---|
| 1/1 | 32 | 6.2 | 6.3 | 9.2 | 7.4 |
| 1/2 | 32 | 21.6 | 20.6 | 19.7 | 17.8 |
| 1/4 | 32 | 27.6 | 27.0 | 26.0 | 24.1 |
| 1/8 | 16 | 15.2 | 15.6 | 14.9 | 14.7 |

steps. In contrast, introducing edit diffusion substantially improves performance, showing that edit operations provide an efficient mechanism for correcting errors left by coarse parallel generation. We also observe that edit-only diffusion can be competitive in some low-budget cases, since edit operations allow more flexible sequence modifications than mask prediction. However, the balanced mask-then-edit design remains more stable across tasks, especially when the initial draft quality is important. These results support our design choice of using mask diffusion for coarse generation and edit diffusion for refinement, rather than relying solely on either component. We provide additional step-allocation analyses under larger generation budgets in Appendix C.4.

**Convergence Behavior of Edit Diffusion.** To analyze how many edit diffusion steps are effectively required during generation, we measure the number of edit steps until the predicted action first becomes empty under greedy decoding. This metric reflects how many refinement steps are actually used to modify the sequence, as subsequent steps no longer change the output once the empty action is predicted.

As shown in Table 3, we observe a clear and consistent trend across datasets: as more steps are allocated to mask diffusion (*i.e.,* under larger generation budgets), the number of required edit steps decreases. This behavior indicates that higher-quality coarse sequences produced by mask diffusion require fewer corrective edits. Importantly, the observed reduction suggests that edit diffusion primarily serves to correct localized inconsistencies rather than to restructure the entire sequence. In practice, only a limited number of edit steps are needed, supporting the view that edit-based refinement acts as a convergent correction process rather than an open-ended sequence rewriting mechanism.

## 5 Related Work

**Masked Diffusion Language Models.** Masked diffusion language models formulate text generation as a discrete denoising process that iteratively refines sequences by predicting masked tokens in arbitrary order (Sahoo et al., 2024), enabling parallel decoding in contrast to autoregressive gen-

eration. Early work on absorbing discrete diffusion established the theoretical foundation for this paradigm, showing that denoising objectives can recover high-quality conditional distributions (Ou et al., 2024). Recent studies demonstrate that masked diffusion models can be effectively scaled to large language modeling settings (Nie et al., 2024), with LLaDA achieving competitive performance at the billion-parameter scale after instruction tuning (Nie et al., 2025). Subsequent work, including Dream, further explores architectural and training refinements and extends masked diffusion to applications such as code generation and infilling (Ye et al., 2025). These results establish MDLMs as a scalable alternative to autoregressive models, while also exposing limitations in fixed-length generation and revision capability that motivate further improvements.

**Enhancing Masked Diffusion Language Models.** Building on masked diffusion, recent work addresses its limitations through soft masks (Hersche et al., 2025), insertions (Kim et al., 2025), and edit-based generation. Some approaches enable variable-length generation by inserting new [MASK] tokens, as in FlexMDM, which supports flexible-length and any-order generation (Kim et al., 2025). Other methods model generation as edit operations (Gu et al., 2019; Reid et al., 2022; Havasi et al., 2025); notably, any-process generation extends MDM with remasking, insertion, and deletion, enabling self-correction, length-variable editing, and adaptive parallelism (Yang et al., 2025b). In parallel, remasking approaches improve revision by re-masking tokens to prevent early commitments (Wang et al., 2025; Huang et al., 2025), while CDLM proposes correction-oriented objectives to better identify erroneous tokens (Zhang et al., 2025). Architectural refinements, such as soft-masked diffusion (Hersche et al., 2025; Zhu et al., 2025; Zhong et al., 2026), replace hard masks with soft token mixtures to improve denoising stability and efficiency. In contrast, our work explicitly targets parallel masked diffusion under a fixed diffusion budget. This requires improving sequence quality without increasing the number of decoding steps. Our edit-based formulation is motivated by structural errors introduced by parallel decoding, such as missing tokens and misaligned spans. Such errors are difficult to handle efficiently with simple re-masking or local regeneration, since insertion or deletion may require regenerating subsequent positions. ME-DLM instead performs localized replacement, deletion, and insertion operations conditioned on the full sequence, enabling targeted refinement while preserving efficiency.

**Coarse-to-refine Paradigms.** Our method is related to coarse-to-refine generation, where an initial draft is produced and then refined. Prior work has explored this idea in autoregressive decoding, such as speculative decoding and draft-then-verify methods (Zhang et al., 2024; Chen et al., 2023a), as well as multi-token verification frameworks such as Medusa (Cai et al., 2024). Iterative refinement has also been studied in language agents, where models improve their own outputs through feedback (Madaan et al., 2023; Shinn et al., 2023). Different from these works, we focus on parallel masked diffusion language models. In this setting, the main challenge is the mismatch between marginal token prediction and sequence-level consistency under parallel decoding. ME-DLM addresses this issue with edit-based refinement under a fixed diffusion budget, using minimal replacement, deletion, and insertion operations to improve the initial masked-diffusion draft.

## 6 Conclusion

We identify a key limitation of masked diffusion language models in parallel multi-token generation, where token-level training objectives fail to adequately capture joint sequence-level consistency. To address this limitation, we propose a simple edit-based refinement framework that corrects residual inconsistencies through minimal, sequence-level edits, effectively approximating sequence-level consistency through iterative correction under aggressive parallel decoding while preserving the efficiency benefits of diffusion-based decoding. Extensive experiments validate the effectiveness of our approach. When built upon LLaDA, ME-DLM achieves gains of 11.6 points on HumanEval and 33.6 points on GSM8K while using only one-eighth of the diffusion steps. An interesting direction for future work is to explore training-free or lightweight mechanisms for enforcing sequence-level constraints during parallel decoding.

## Impact Statement

This paper presents a technical study of masked diffusion language models under parallel decoding and proposes an edit-based refinement mechanism to improve sequence-level consistency. The contribution is methodological in nature and primarily aims to better understand and address a limitation in existing generative models. The proposed method does not introduce new application domains or capabilities beyond improving generation quality and efficiency. As such, we do not anticipate specific societal impacts beyond those generally associated with large language models.

## Acknowledgment

This project is funded in part by Shenzhen Loop Area Institute, by the Centre for Perceptual and Interactive Intelligence (CPII) Ltd under the Innovation and Technology Commission (ITC)'s InnoHK, in part by NSFC-RGC Project N_CUHK498/24, and in part by Guangdong Basic and Applied Basic Research Foundation (No. 2023B1515130008, XW).

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

# Appendix

## A    Inference code

The following code snippet illustrates the inference procedure of our edit-based refinement. All edit operations, including replacement, deletion, and insertion, are applied fully in parallel across token positions. As a result, the refinement step introduces only negligible computational overhead compared to a standard parallel unmasking step.

```python
curr_ids, next_ids = model(input_ids)
next_ids = next_ids.roll(1)
curr_ids[prompt_mask] = next_ids[prompt_mask] = input_ids[prompt_mask]

# Replace Tokens
input_ids = curr_ids

# Delete Tokens
mask = torch.ne(input_ids, del_id)
input_ids, next_ids = input_ids[mask], next_ids[mask]

# Insert Tokens
ins_mask = torch.ne(input_ids, next_ids)
all_ids = torch.stack([next_ids, input_ids], dim=1).view(-1)
all_mask = torch.stack([ins_mask, torch.ones_like(input_ids).bool()], dim=1).view(-1)
input_ids = all_ids[all_mask]
```

## B    Inference Time

We measure inference efficiency on NVIDIA H800 GPUs. To fully utilize GPU resources, multiple cases are concatenated into a single batch during inference. Variable-length sequences are processed using "flash_attn_varlen_func" from Flash-Attention (Dao, 2023), which enables efficient attention computation without padding overhead. The reported inference time per case is computed by dividing the total wall-clock inference time by the number of cases processed in the batch.

Table 4 reports inference efficiency under different allocations between mask diffusion and edit diffusion while keeping the total number of diffusion steps fixed. Across both high-budget (512 steps) and low-budget (64 steps) settings, introducing edit diffusion does not lead to increased inference latency. The inference time per case remains highly stable across different allocations, with only minor fluctuations on the order of 0.01 seconds as the number of edit steps increases. Correspondingly, throughput measured in tokens per second also remains nearly constant. These results indicate that the proposed edit-based refinement introduces negligible computational overhead and preserves the efficiency of parallel diffusion decoding.

Under the more aggressive low-budget setting with only 64 total diffusion steps, we further observe that inference time not only remains stable but, in some cases, slightly decreases as more steps are allocated to edit diffusion. We attribute this behavior to differences in token selection and scheduling between the two diffusion phases. Specifically, mask diffusion requires selecting subsets of masked tokens based on confidence scores within each individual sequence, which involves per-sequence operations and synchronization. In contrast, edit diffusion applies edit actions fully in parallel over concatenated sequences, without requiring per-sequence confidence-based token selection. As a result, edit diffusion can more effectively exploit batched execution over concatenated cases, leading to marginally lower inference time in low-step regimes.

*Table 4.* Inference time under different allocations of mask diffusion and edit diffusion steps. Experiments are conducted on NVIDIA H800 GPUs under two total diffusion budgets (512 and 64 steps). For each budget, we vary the allocation between mask diffusion and edit diffusion while keeping the total number of steps fixed. Reported inference time is averaged per case, and throughput is measured in tokens per second.

| | Total 512 diffusion steps | | | | | Total 64 diffusion steps | | | | |
|---|---|---|---|---|---|---|---|---|---|---|
| Mask Step | 512 | 504 | 496 | 488 | 480 | 64 | 56 | 48 | 40 | 32 |
| Edit Step | 0 | 8 | 16 | 24 | 32 | 0 | 8 | 16 | 24 | 32 |
| Avg. inference time per case (sec) | 12.13 | 12.12 | 12.16 | 12.15 | 12.10 | 1.53 | 1.53 | 1.50 | 1.48 | 1.47 |
| Avg. tokens per second | 42.20 | 42.24 | 42.11 | 42.14 | 42.31 | 334.64 | 334.64 | 341.33 | 345.94 | 348.30 |

*Table 5.* General evaluation results on MMLU, ARC-C, and TruthfulQA under different decoding budgets. All results are evaluated under a fully generative protocol. The best result under each budget and benchmark is highlighted in bold.

| Methods | Budget | Evaluation Dataset | | | Budget | Evaluation Dataset | | |
|---------|--------|------|-------|-----------|--------|------|-------|-----------|
| | | MMLU | ARC-C | TruthfulQA | | MMLU | ARC-C | TruthfulQA |
| LLaDA | | 60.2 | **82.2** | **56.8** | | 57.1 | 77.2 | 49.8 |
| ME-DLM Stage 2 | 1/1 | 60.6 | 79.7 | 52.5 | 1/2 | 57.7 | 74.4 | 48.6 |
| ME-DLM Stage 3 | | **62.6** | 81.1 | 53.5 | | **61.5** | **80.7** | **54.6** |
| LLaDA | | 44.5 | 60.7 | 44.4 | | 23.4 | 29.9 | 23.3 |
| ME-DLM Stage 2 | 1/4 | 43.3 | 58.4 | 36.1 | 1/8 | 21.7 | 26.0 | 19.5 |
| ME-DLM Stage 3 | | **58.5** | **73.6** | **49.8** | | **56.2** | **69.2** | **46.9** |

*Table 6.* Ablation study on refinement-state construction for edit-based training. We compare the full Stage-3 training with two lower-cost alternatives: using only mask-diffusion states and using rule-based noisy sequences. The best result under each budget and benchmark is highlighted in bold.

| Budget | Model | Evaluation Dataset | | | | | | Average |
|--------|-------|-----------|------------|------|-------|-------|----------|---------|
| | | HumanEval | HumanEval+ | MBPP | MBPP+ | GSM8K | MATH-500 | |
| 1/1 | ME-DLM Stage-2 | 56.1 | 50.0 | 54.8 | 46.8 | 80.9 | 45.8 | 55.7 |
| | ME-DLM Stage-3 | **57.9** | **53.0** | **61.6** | **52.9** | **84.8** | **50.0** | **60.0** |
| | w/ mask-only states | 52.4 | 48.8 | 57.9 | 51.6 | 84.2 | 49.4 | 57.4 |
| | w/ rule-based noise | 54.3 | 50.0 | 56.3 | 45.8 | 78.8 | 44.8 | 55.0 |
| 1/2 | ME-DLM Stage-2 | 45.7 | 42.7 | 52.4 | 44.2 | 76.3 | 43.0 | 50.7 |
| | ME-DLM Stage-3 | **51.2** | **46.3** | **54.8** | **47.6** | **82.7** | **50.0** | **55.4** |
| | w/ mask-only states | 48.8 | 43.9 | 48.7 | 41.5 | 81.1 | 49.0 | 52.2 |
| | w/ rule-based noise | 47.0 | 43.9 | 53.7 | 44.4 | 72.6 | 40.8 | 50.4 |
| 1/4 | ME-DLM Stage-2 | 36.6 | 32.9 | 34.7 | 31.0 | 58.9 | 32.0 | 37.7 |
| | ME-DLM Stage-3 | **42.1** | **39.0** | **45.0** | **37.6** | **76.6** | 38.2 | **46.4** |
| | w/ mask-only states | 38.4 | 35.4 | 33.9 | 29.4 | 74.1 | **40.4** | 41.9 |
| | w/ rule-based noise | 39.0 | 36.6 | 42.9 | 35.4 | 55.6 | 32.6 | 40.4 |
| 1/8 | ME-DLM Stage-2 | 13.4 | 13.4 | 22.0 | 18.8 | 30.2 | 17.8 | 19.3 |
| | ME-DLM Stage-3 | **25.0** | **22.6** | **26.7** | 22.8 | **63.8** | **34.4** | **32.6** |
| | w/ mask-only states | **25.0** | **22.6** | 25.9 | **24.1** | 61.4 | 31.0 | 31.7 |
| | w/ rule-based noise | 21.3 | 20.1 | 21.2 | 18.3 | 29.6 | 18.0 | 21.4 |

# C  Additional Experiments

## C.1  Generalization Evaluation

To further examine whether the proposed edit-based refinement generalizes beyond code generation and mathematical reasoning, we evaluate ME-DLM on three general benchmarks: MMLU (Hendrycks et al., 2020), ARC-C (Clark et al., 2018), and TruthfulQA (Lin et al., 2022). We adopt a fully generative evaluation protocol, where the model is required to directly generate the final answer in a \boxed{} format. This protocol is stricter than standard multiple-choice evaluation based on option likelihood, and therefore the absolute numbers are not directly comparable to those reported under likelihood-based evaluation settings.

As shown in Table 5, ME-DLM Stage 3 consistently improves over ME-DLM Stage 2 across all decoding budgets and all three benchmarks. The gains become especially pronounced under aggressive parallel decoding. For example, at the $1/8$ budget, Stage 3 improves MMLU from 21.7 to 56.2, ARC-C from 26.0 to 69.2, and TruthfulQA from 19.5 to 46.9. These results indicate that edit-based refinement effectively mitigates the quality degradation caused by highly parallel masked diffusion decoding, even on general knowledge and commonsense reasoning tasks. We also observe that ME-DLM Stage 2 sometimes underperforms the original LLaDA baseline, particularly on ARC-C and TruthfulQA. This is likely due to

*Table 7.* Parameter analysis results. Results are reported as pass@1 for code tasks and accuracy for math tasks. Average denotes the mean score over all evaluation benchmarks.

| Budget | Variant | Evaluation Dataset | | | | | | Average |
|---|---|---|---|---|---|---|---|---|
| | | HumanEval | HumanEval+ | MBPP | MBPP+ | GSM8K | MATH-500 | |
| 1/1 | $\alpha = 0.5, \beta = 0$ | 57.9 | 53.0 | **61.6** | **52.9** | 84.8 | 50.0 | **60.0** |
| | $\alpha = 0.25, \beta = 0$ | 56.7 | 51.8 | 60.3 | 51.3 | 84.8 | 50.2 | 59.2 |
| | $\alpha = 0.75, \beta = 0$ | **59.8** | **54.9** | 57.9 | 49.7 | 84.4 | 51.0 | 59.6 |
| | $\alpha = 0.5, \beta = 0.5$ | 51.8 | 47.6 | 57.1 | 48.4 | **85.6** | **51.4** | 57.0 |
| 1/2 | $\alpha = 0.5, \beta = 0$ | 51.2 | 46.3 | **54.8** | **47.6** | 82.7 | **50.0** | 55.4 |
| | $\alpha = 0.25, \beta = 0$ | 49.4 | 45.7 | 50.3 | 42.6 | 81.7 | 46.0 | 52.6 |
| | $\alpha = 0.75, \beta = 0$ | 47.6 | 44.5 | 53.2 | 46.3 | 82.7 | 46.0 | 53.4 |
| | $\alpha = 0.5, \beta = 0.5$ | **54.9** | **50.6** | 54.5 | 47.6 | **85.1** | 49.0 | **56.9** |
| 1/4 | $\alpha = 0.5, \beta = 0$ | 42.1 | 39.0 | 45.0 | 37.6 | 76.6 | 38.2 | 46.4 |
| | $\alpha = 0.25, \beta = 0$ | 40.9 | 37.2 | 39.9 | 34.7 | 74.1 | 42.8 | 44.9 |
| | $\alpha = 0.75, \beta = 0$ | 39.6 | 34.8 | 45.2 | 39.2 | 78.8 | 41.4 | 46.5 |
| | $\alpha = 0.5, \beta = 0.5$ | **43.3** | **40.2** | **48.1** | **40.7** | **80.7** | **44.4** | **49.6** |
| 1/8 | $\alpha = 0.5, \beta = 0$ | 25.0 | 22.6 | 26.7 | 22.8 | 63.8 | 34.4 | 32.6 |
| | $\alpha = 0.25, \beta = 0$ | 27.4 | 26.2 | 28.3 | 24.3 | 60.0 | 34.8 | 33.5 |
| | $\alpha = 0.75, \beta = 0$ | 25.6 | 23.8 | 31.7 | 30.4 | 64.7 | 34.4 | 35.1 |
| | $\alpha = 0.5, \beta = 0.5$ | **31.7** | **29.3** | **34.7** | **29.6** | **71.6** | **37.8** | **39.1** |

the limited coverage of general-purpose instruction data during our supervised fine-tuning stage. Nevertheless, Stage 3 substantially reduces or reverses this gap in most settings, suggesting that edit-based refinement improves robustness even when the initial masked-diffusion model is affected by suboptimal training data coverage. Overall, these results demonstrate that the proposed refinement mechanism is not limited to code or mathematical reasoning, but also provides strong benefits on broader general-purpose evaluation tasks.

## C.2   Ablation on Refinement-State Construction

We further study the cost of constructing refinement states during Stage-3 training. In the full setting, edit-refinement supervision is built from model-generated intermediate states, which better match the inference-time distribution but introduce additional training overhead. To evaluate whether this cost is necessary, we compare with two cheaper alternatives: *mask-only states*, where edit rollout is removed during training, and *rule-based noise*, where noisy inputs are constructed by random insertion, deletion, and replacement without model rollout.

As shown in Table 6, the full Stage-3 training achieves the best average performance across all budgets. Although mask-only states reduce the cost of collecting refinement states, they consistently underperform the full Stage-3 model on average. The gap is especially clear on code benchmarks, where outputs are sensitive to small token-level errors. Rule-based noise performs even worse, particularly on mathematical reasoning tasks, suggesting that randomly corrupted sequences do not match the error distribution produced by parallel diffusion decoding. These results indicate that the main challenge is not merely reducing the cost of refinement-state construction, but obtaining training states that resemble test-time model errors. Model-generated refinement states provide more effective supervision for edit-based correction, leading to better robustness under aggressive parallel decoding.

## C.3   Parameter Analysis

Here, we analyze the sensitivity of our method to two key hyperparameters in Stage-3 training. The first parameter is $\alpha$, which controls the proportion of mask-edit diffusion training relative to standard mask diffusion. In the main experiments, we set $\alpha = 0.5$, allocating equal training budget to mask diffusion and edit-based refinement. The second parameter is the noise range used for edit diffusion, denoted by $\beta$, which specifies the minimum diffusion time $t$ from which edit refinement samples are drawn. By default, we set $\beta = 0$, *i.e.,* $t \in (0, 1]$, allowing edit diffusion to operate on sequences corrupted by arbitrary noise levels. In this subsection, we vary $\alpha \in \{0.25, 0.5, 0.75\}$ and consider a restricted edit diffusion range with

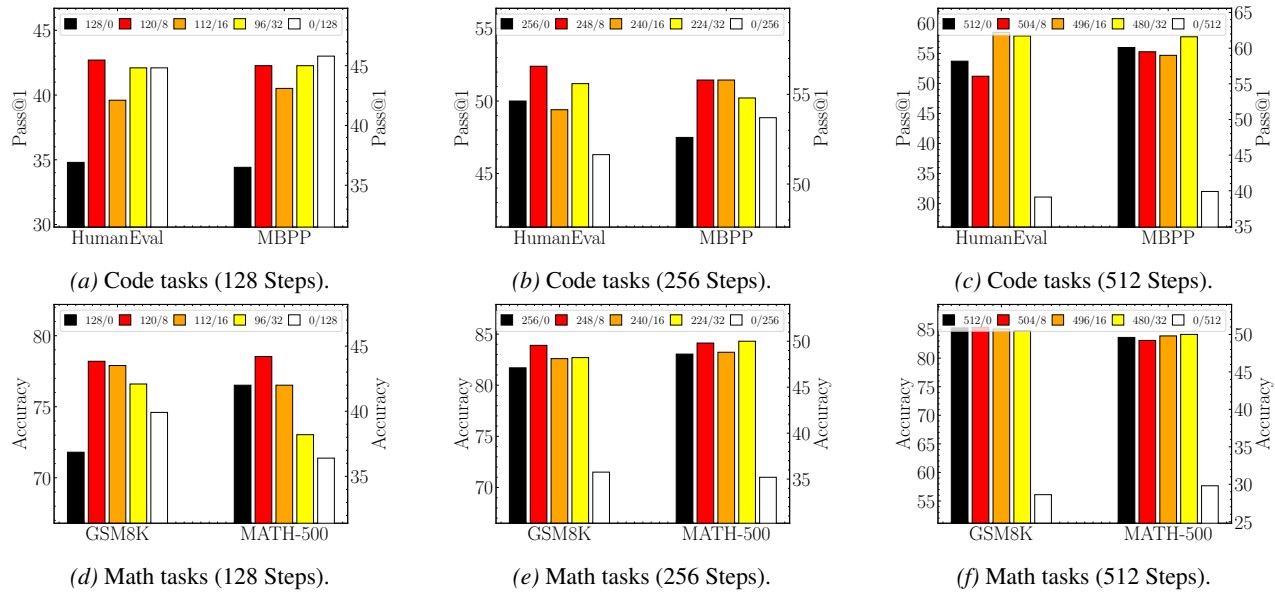

*Figure 4.* Effect of different allocations between mask diffusion and edit diffusion under three generation budgets. Each legend entry $m/e$ indicates the number of mask diffusion steps and edit diffusion steps, respectively.

$\beta = 0.5$, and evaluate their effects under different generation budgets.

The results are summarized in Table 7. Under the full budget setting (1/1), different parameter configurations lead to relatively small performance differences. Most of the observed variations can be attributed to statistical fluctuation rather than systematic trends. In particular, the setting with $\beta = 0.5$ shows slightly worse and more unstable results on HumanEval, which is expected given the relatively small size of this benchmark and the higher variance induced by restricting the edit diffusion noise range.

When the generation budget is reduced (1/4 and 1/8), the advantage of setting $\beta = 0.5$ becomes much more pronounced. In these low-budget regimes, restricting edit diffusion to intermediate noise levels consistently yields clear performance improvements across both code and math tasks. This is because sampling edit diffusion steps from a narrower noise range effectively increases the difficulty of the refinement task, forcing the model to learn stronger and more targeted correction behaviors, which are particularly beneficial when aggressive parallel decoding introduces larger inconsistencies.

In contrast, the impact of $\alpha$ is relatively mild across all budgets. Varying the proportion between mask diffusion and edit diffusion does not significantly change the final performance. This can be explained by a trade-off effect between the two components. When $\alpha$ is smaller, the model benefits from stronger mask diffusion capability, but the edit-based refinement becomes weaker; when $\alpha$ is larger, edit refinement improves at the cost of mask diffusion quality. Since both mask diffusion and edit diffusion contribute to the final generation quality at inference time, these effects largely offset each other, resulting in comparable overall performance across different $\alpha$ values.

## C.4 Effect of Step Allocation

To further examine the interaction between mask diffusion and edit diffusion, we extend the step-allocation analysis to three total generation budgets: 128, 256, and 512 diffusion steps. For each budget, we first consider a set of mixed allocations in which the number of edit diffusion steps is at most 32, matching the training configuration. Within this range, we gradually reduce the number of edit steps while keeping the total number of diffusion steps fixed, allowing us to evaluate how different levels of edit-based refinement affect generation quality under varying budget regimes. In addition, we include an extreme edit-only setting, where all diffusion steps are assigned to edit diffusion and the number of mask diffusion steps is set to zero. This additional setting allows us to test whether edit diffusion can function effectively as a standalone generation mechanism, beyond its intended role as a refinement module.

As shown in Figure 4, several consistent trends emerge. First, the benefit of allocating part of the generation budget to edit diffusion is more pronounced under smaller budgets, where parallel unmasking alone is more likely to introduce structural

| *Mask Diffusion* | *Mask-Edit Diffusion* |
|---|---|

```
def recursive_list_sum(lst):
    ### =

    for item in lst

        if isinstance(item, list):
            total += recursive_list_sum(item)
        else:
            total += item
    return total
```

```
def recursive_list_sum(lst):
    total = 0

    for item in lst:

        if isinstance(item, list):
            total += recursive_list_sum(item)
        else:
            total += item
    return total
```

*Figure 5.* Comparison between mask diffusion and mask-edit diffusion on a code example.

| *Mask Diffusion* | *Mask-Edit Diffusion* |
|---|---|

**Mask Diffusion:**

Sylvie's monthly salary is initially \$600, so her annual salary is \$600 \times 12 = *\$7200\$*

The company has a policy of increasing salaries by 10% of the initial salary every year for employees who have been in the company for five least. years. Sylvie has been in the company for five years, so she qualifies for annual increases.

- After the first increase (at the end of the 6th *year:*
Increase = 10% of \$7200 = 0.10 \times 7200 = 720\$
New salary = \$7200 + 720 = 7920\$

- After the second increase (at the end of the 7th year):
Increase = 10% of \$7200 = 0.10 \times 7200 = 720\$
New salary = \$7920 + 720 = 8640\$

- After the third increase (at the end of the 8th year):
Increase = 10% of \$7200 = 0.10 \times 7200 = 720\$
New salary = \$8640 + 720 = 9360\$

*S,* after three more years of service (i.e., after 8 *years year)*, her annual salary is \$9360.

Alternatively, using the formula for compound growth:
*- salary* after 3 increases = \$7200 \times (1 + 0.10)^3 = 7200 \times (1.10)^3\$
\$(1.10)^3 = 1.10 \times 1.10 \times 1.10 = 1.331\$
\$7200 \times 1.331 = 9360\$

Both methods confirm that the annual salary is \$9360.

*\boxed9960}*<|im_end|>

**Mask-Edit Diffusion:**

Sylvie's monthly salary is initially \$600, so her annual salary is \$600 \times 12 = *7200\$.*

The company has a policy of increasing salaries by 10% of the initial salary every year for employees who have been in the company for five years. Sylvie has been in the company for five years, so she qualifies for three annual increases.

- After the first increase (at the end of the 6th *year):*
Increase = 10% of \$7200 = 0.10 \times 7200 = 720\$
New salary = \$7200 + 720 = 7920\$

- After the second increase (at the end of the 7th year):
Increase = 10% of \$7200 = 0.10 \times 7200 = 720\$
New salary = \$7920 + 720 = 8640\$

- After the third increase (at the end of the 8th year):
Increase = 10% of \$7200 = 0.10 \times 7200 = 720\$
New salary = \$8640 + 720 = 9360\$

*Thus,* after three more years of service (i.e., after 8 *years)*, her annual salary is \$9360.

Alternatively, using the formula for compound growth:
*Salary* after 3 increases = \$7200 \times (1 + 0.10)^3 = 7200 \times (1.10)^3\$
\$(1.10)^3 = 1.10 \times 1.10 \times 1.10 = 1.331\$
\$7200 \times 1.331 = 9360\$

Both methods confirm that the annual salary is \$9360.

*\boxed{9360}*<|im_end|>

*Figure 6.* Comparison between mask diffusion and mask-edit diffusion on a math example.

inconsistencies. As the total number of diffusion steps increases, the performance gap between different mixed allocations becomes smaller, suggesting that additional mask diffusion steps can partially compensate for the absence of stronger refinement. Second, incorporating edit diffusion during training can improve performance even when edit diffusion is used only lightly at inference time. In particular, for math reasoning tasks under a budget of 512 steps, configurations trained with edit-based refinement outperform pure mask diffusion baselines. For example, on MATH-500, performance improves from 45.8 to 49.6 at 512 steps. This observation is consistent with prior findings that structured refinement objectives can improve the quality of parallel generation even when the refinement mechanism is not heavily used during inference (Song et al., 2025). Finally, the edit-only setting generally underperforms the proposed mixed mask-edit strategy, and this gap becomes more evident at larger budgets. We attribute this to a distribution mismatch: during training, edit operations are applied to sequences that have already been partially organized by mask diffusion, whereas starting from edit diffusion alone produces intermediate states that deviate substantially from the training distribution. Together, these results support our design choice of combining mask diffusion for coarse generation with edit diffusion for refinement, showing that edit diffusion serves best as a complementary mechanism that improves both low-budget robustness and high-budget generation quality.

## C.5 Case Study

We provide qualitative case studies on code generation tasks and math reasoning tasks to illustrate how mask-edit diffusion improves over mask diffusion under parallel multi-token decoding. For all cases, we use the same prompt and the same

mask diffusion schedule to obtain an initial complete output. Mask-edit diffusion then performs a lightweight refinement phase conditioned on the full generated sequence, applying a small set of localized edit actions to restore global consistency.

In the code example (Figure 5), mask diffusion successfully captures the intended recursive structure but introduces control-flow corruption, including missing punctuation and structure tokens (*e.g.,* the colon in `for item in lst:`), missing variable initialization (*e.g.,* `total = 0`), and malformed branching syntax. While each local fragment resembles valid Python, these errors jointly break the block structure and render the program non-executable. Mask-edit diffusion corrects these issues through a small set of targeted edits, restoring indentation-consistent blocks and a valid recursive implementation. In the math example (Figure 6), mask diffusion samples the final answer early (via confidence-based selection) before generating the full derivation. As a result, the model commits to an under-supported final span and produces a wrong answer (9960) at the end, yielding an incorrect reported answer even though the subsequent reasoning correctly derives 9360. Mask-edit diffusion fixes this by performing a localized replacement on the final-answer span, aligning the final output with the already-correct computation.

