# OpenReview forum: "Edit-Based Refinement for Parallel Masked Diffusion Language Models"
_ICML.cc/2026/Conference — ICML 2026 regular_

### Official Review · Reviewer_1Eqa · 2026-03-10

**Soundness:** 3
**Presentation:** 3
**Significance:** 3
**Originality:** 2
**Overall Recommendation:** 4
**Confidence:** 4

**Summary:**

This paper identifies a critical limitation in masked diffusion language models (MDLMs): when generating multiple tokens in parallel, these models suffer from degraded quality due to a mismatch between token-level training objectives and the need for joint sequence consistency. The authors propose ME-DLM (Mask-Edit Diffusion Language Model), a two-stage framework that augments standard masked diffusion with lightweight edit-based refinement. After producing an initial sequence through parallel unmasking, the model performs iterative refinement using replacement, deletion, and insertion operations conditioned on the full sequence. Training supervision is derived from edit distance, providing deterministic signals for learning minimal corrections. Built upon LLaDA-8B, ME-DLM achieves substantial improvements on code generation (HumanEval: +11.6%) and mathematical reasoning (GSM8K: +33.6%) under aggressive parallel decoding with only 1/8 of diffusion steps, while introducing negligible computational overhead.

**Compliance With Llm Reviewing Policy:**

Affirmed.

**Final Justification:**

Please see the rebuttal acknowledgement.

**Key Questions For Authors:**

NA

**Strengths And Weaknesses:**

Strengths:
1.The paper is generally well-structured, following a logical progression from problem identification to method, to experiments.
2.The gains are practically significant, with 33.6% improvement on GSM8K and 11.6% on HumanEval at 1/8 budget represent substantial advances that could enable practical deployment of diffusion LMs with aggressive parallelism.
3.Parallel decoding efficiency is a critical bottleneck for diffusion language models competing with autoregressive approaches. The identified failure mode (marginal vs. joint consistency) is fundamental and likely affects other parallel generation paradigms.

Weaknesses:
1.The evaluation is limited to code generation and mathematical reasoning tasks. While these are important reasoning benchmarks, broader evaluation on open-ended generation, summarization, or dialogue would strengthen claims about general applicability.
2.The two-stage paradigm (coarse generation + refinement) has parallels in other areas (e.g., draft-then-refine in autoregressive models, or cascaded diffusion in continuous domains). The novelty lies in the specific instantiation for masked diffusion LMs, but this connection is not discussed.
3.The concurrent work on corrective diffusion (Zhang et al., 2025) and remasking-based approaches (Wang et al., 2025; Huang et al., 2025) address similar problems. The distinction from these concurrent approaches is mentioned but could be more thoroughly developed.

---

> ### Author Rebuttal · Authors · 2026-03-29
>
> > W1: The evaluation is limited to code generation and mathematical reasoning tasks.
>
> We thank the reviewer for pointing out the limitation of our evaluation scope. To assess generalization beyond code and mathematical reasoning, we further evaluate our method on MMLU, ARC-C, and TruthfulQA.
>
> |Budget|Model|MMLU|ARC-C|TruthfulQA|
> |:-:|:-:|:-:|:-:|:-:|
> |1/1|LLaDA|60.2|**82.2**|**56.8**|
> |1/1|ME-DLM Stage 2|60.6|79.7|52.5|
> |1/1|ME-DLM Stage 3|**62.6**|81.1|53.5|
> |1/2|LLaDA|57.1|77.2|49.8|
> |1/2|ME-DLM Stage 2|57.7|74.4|48.6|
> |1/2|ME-DLM Stage 3|**61.5**|**80.7**|**54.6**|
> |1/4|LLaDA|44.5|60.7|44.4|
> |1/4|ME-DLM Stage 2|43.3|58.4|36.1|
> |1/4|ME-DLM Stage 3|**58.5**|**73.6**|**49.8**|
> |1/8|LLaDA|23.4|29.9|23.3|
> |1/8|ME-DLM Stage 2|21.7|26.0|19.5|
> |1/8|ME-DLM Stage 3|**56.2**|**69.2**|**46.9**|
>
> Note: We adopt a fully generative evaluation protocol, requiring the model to produce the final answer in a \boxed{} format. This setting is stricter than standard multiple-choice evaluation based on option likelihood, and thus results are not directly comparable to LLaDA paper.
>
> From the results, we observe several consistent trends:
> - Stage-3 consistently outperforms Stage-2 across all budgets, and the improvement becomes more pronounced under lower budgets (e.g., 1/4 and 1/8). This shows that edit-based refinement effectively mitigates errors introduced by aggressive parallel decoding, and generalizes beyond reasoning tasks to broader knowledge benchmarks.
> Robustness to SFT data imbalance.
> - Due to limited coverage of general-purpose instruction data in our SFT stage, Stage-2 may underperform LLaDA on some benchmarks (e.g., ARC-C, TruthfulQA). However, Stage-3 significantly reduces or even reverses this gap, indicating that our method improves generalization even under suboptimal data conditions.
>
> > W2: The novelty lies in the specific instantiation for masked diffusion LMs, but this connection is not discussed.
>
> We thank the reviewer for highlighting the connection to broader coarse-to-refine paradigms. Indeed, similar two-stage generation strategies have been explored across multiple settings.
>
> Prior work has studied draft-then-verify schemes such as speculative decoding [1–3], where a coarse sequence is first generated by a lightweight model and then verified or corrected by a stronger model. In addition, in the context of language agents, iterative refinement with self-feedback has been explored in works [4-5], where models generate an initial draft and iteratively improve it based on feedback.
>
> However, our work differs from these prior approaches in several key aspects.
> - We study this paradigm specifically in the context of parallel masked diffusion language models, where the main challenge arises from the mismatch between marginal token prediction and sequence-level consistency.
> - Our method is designed to operate under a fixed diffusion budget, explicitly optimizing the quality–efficiency trade-off.
>
> We will revise the paper to better position our work within this line of research and clarify these connections.
>
> [1] Accelerating Large Language Model Decoding with Speculative Sampling
>
> [2] Draft & Verify: Lossless Large Language Model Acceleration via Self-Speculative Decoding
>
> [3] Medusa: Simple LLM Inference Acceleration Framework with Multiple Decoding Heads
>
> [4] Reflexion: Language Agents with Verbal Reinforcement Learning
>
> [5] Self-Refine: Iterative Refinement with Self-Feedback
>
> > W3: The distinction from these concurrent approaches is mentioned but could be more thoroughly developed.
>
> We thank the reviewer for pointing out the connection to concurrent corrective diffusion and remasking-based approaches. We would like to clarify that our work differs from these approaches in a fundamental aspect: we explicitly study and optimize parallel masked diffusion under a fixed diffusion budget.
>
> Specifically, our goal is to improve generation quality under a constrained number of diffusion steps, which directly corresponds to practical inference efficiency. In contrast, prior approaches such as corrective diffusion and remasking-based methods primarily focus on whether errors made during decoding can be corrected through additional refinement steps. These methods typically do not treat the diffusion budget as a primary constraint, and in many cases introduce extra decoding iterations or adaptive refinement procedures, leading to increased inference cost.
>
> From a methodological perspective, the training objective and design principles are different. Our method is trained to perform minimal, globally conditioned edits that improve sequence-level consistency within a fixed step budget, whereas prior work is generally optimized for error detection and correction during iterative decoding, without explicitly enforcing efficiency constraints.
>
> We will revise the paper to make this distinction clearer in the related work section.

---

> > ### Author Rebuttal · Reviewer_1Eqa · 2026-04-05
> >
> > NA

---

### Official Review · Reviewer_LtYi · 2026-03-11

**Soundness:** 2
**Presentation:** 2
**Significance:** 3
**Originality:** 3
**Overall Recommendation:** 2
**Confidence:** 4

**Summary:**

The paper proposed to train a separate parallel edit-based refinement model on top of the output of a masked diffusion language model (MDLM).

The refinement model gets as input the output after the MDLM inference, so what the MDLM predicts as final seq. Then, for every input token in the input seq, it predicts two outputs c and n. c can be [del], allowing for deletions, or some other token, which could be the input, or a different one, allowing for substitutions. n can be the same as c, in which case it is ignored, or it can be a different token, in which case it would be a new insertion.

The inference first uses the MDLM to produce a sequence via the standard diffusion inference iteration steps, and then the separate refinement model is run on top of that for multiple iteration steps.

**Compliance With Llm Reviewing Policy:**

Affirmed.

**Final Justification:**

The rebuttal did not change my evaluation.

**Key Questions For Authors:**

"lightweight post-editing steps" - what does lightweight mean here? Is this really more lightweight than the MDLM?

"discrepancy between marginal token prediction and joint sequence validity" - I think I understand the argument, but can this be made more specific? Can this be measured? Can this be mathematically formulated?

lightweight refinement after parallel multi-token generation: editing diffusion steps. These are parallel as well? Wouldn't that really have the same issue?

Eval on HumanEval and GSM8K. Should there be other evals?  Zero-shot perplexities on Wikitext et al? GLUE evaluation? Gen. PPL?

Figure 2 is not really clear. Where do I see the edits there? What are those n? What does the equal sign mean here? After reading the text, it somewhat becomes clear, but I think such a figure would be much more useful when you can understand it without reading the text.

Sec 4.2, the n should also allow to be sth like [empty], to not insert sth? Ok, it is just by checking c != n. But so that means the model cannot insert the same token again?

And there is no way to insert sth at pos 0 then?

Sec 4.3 "Given a ground-truth target sequence x⋆, we first obtain an initial noisy sequence x(0) via random masking and parallel multi-token unmasking." - interesting, but how exactly? The parallel multi-token unmasking, how?

x^(0) is normally complete noise, completely masked? But here it is not, it is only partially masked? Ah, wait, this is only for the refinement model? This confused me initially. But as written before, I think it would be interesting if the refinement model is also trained on completely empty or random seqs, and allow to run only this model.

Inference: So, it is first the normal MDLM, then the edit-based refinement, and then stop. Why not repeat? I guess because the normal MDLM has not really been trained on such seqs. But anyway, this seems a bit arbitrary.

Also, the whole approach sounds just like model combination? Why not use only edit-based refinement as the sole model?

Sec 5.1 "pretrain the model" - which one? The new refinement model?

"enable joint prediction of the current and sub-sequent tokens, which serves as the basis for token-wise edit modeling" - I don't exactly understand. Current and subsequent, is this about c and n?

"proposed unmask-and-edit diffusion training" - I thought the model was proposed, and the training comes with the model? But already before, you were training the refinement model? But then with different training? I'm confused.

"built upon LLaDA-8B" - is that model frozen, or finetuned?

Table 1, can those numbers be compared to Soft Mask or EvoToken? Were those models trained on the same data? If not, the comparison is a bit pointless?

What exactly is ME-DLM Stage-2 and ME-DLM Stage-3?

I don't fully understand the approach. Let's speak about inference. There is the base MDLM (Llada) model, which is run a few times, and then produces some seq. Then the new refinement model runs on it, edits the seq, a few steps, and then that is the output?

When you speak about training, is the base MDLM trained in any way, or frozen the whole time? Only the refinement model is trained?

Code not released?

**Limitations:**

yes

**Strengths And Weaknesses:**

Strengths:

* Supporting edit-based refinement is very reasonable for code generation, and an important approach to explore.
* The results look good.

Weaknesses:

* The complexity of having two models, the MDLM and the refinement model, is quite a bit increased.
* Quite a lot of interesting ablations are missing. E.g. (just to give an example), the edit-based model could be the sole main mode, and the whole process could use only that part. How does it perform then? I understand this is maybe not so trivial and needs to be trained differently than how it is trained right now, e.g. that it could also start from a completely empty seq.
* Some parts of the approach are unclear. E.g. the multi-stage training is not totally clear to me. Using LLaDA-8B as base model is not clear to me, does it mean this is the MDLM model, or the refinement model? See my comments/questions below.
* The comparison to literature in Table 1 is maybe not reasonable due to different training data.
* Code not published?

---

> ### Author Rebuttal · Authors · 2026-03-29
>
> > High-level description of training and inference.
>
> For training, we perform full fine-tuning of all model parameters. And we clarify that our method uses a **single** model, not two separate models. The model is initialized from LLaDA-8B-Base (a pretrained MDLM model, see line 324) and trained through multiple stages with shared parameters. Masked diffusion and edit diffusion differ only in the input sequence (masked vs. complete) and the decoding rule (unmasking vs. editing), not in the network weights.
>
> Training proceeds in stages. In Stage 1/2, the model is trained with masked diffusion only. In addition to predicting the current token c, the model is also trained to predict the next token n at each position. This does not introduce explicit edits, but equips the model to jointly predict the current token and the next token at each position, which later serves as the basis for edit-based refinement. In Stage 3, we jointly train the model for masked diffusion and edit diffusion.
>
> For inference, the model first performs masked diffusion generation, given the instruction and a sequence of mask tokens, it progressively unmasks tokens to produce an initial complete draft. This draft is then further refined via edit-based diffusion. Notably, the number of edit diffusion steps is much smaller than masked diffusion steps (32 vs. 480 when total steps = 512), so edit refinement adds only a small fraction of the total decoding steps.
>
> We provide the training and inference code in the supplementary material for reproducibility.
>
> > Details of Edit.
>
> We do not explicitly introduce an empty token for insertion, as insertions are already fully expressed through the positional $(c_i, n_i)$ formulation. This also avoids introducing a large number of trivial no-op labels in training, which we found unnecessary in practice.
>
> Repeated-token insertion does not rely on inserting a token between identical tokens. Instead, it is realized through insertions at adjacent positions. For example, with a sequence `a a b`, inserting a token `a` between `a` and `a` can be achieved by inserting `a` between `a` and `b`, yielding an equivalent sequence `a a a b`. This behavior follows directly from the definition of the insertion operator over positions, rather than requiring a special case.
>
> Boundary insertion is handled by treating the transition between the prompt and the first generated token as a valid editable position.
>
> We will revise the paper to make these cases explicit and improve Figure 2 to better illustrate how edit operations are applied in practice.
>
> > Details of training.
>
> Stage 1/2 follows standard masked diffusion, sample $t \in [0,1)$, randomly mask tokens according to t, and train the model to predict the masked tokens with a token-level objective.
>
> For Stage 3, the procedure differs in that we simulate the inference process to construct refinement supervision. We first sample $t \in [0,1)$ and obtain a partially masked sequence. The model then performs parallel multi-token unmasking in an inference manner (without supervision). At each step, a subset of masked positions is randomly selected and the model fills these positions with its own predictions, repeating until no masks remain. During training, the number of tokens unmasked per step is randomly chosen from {2, 4, 8, 16}. This produces a noisy complete sequence, from which edit targets are computed against the ground truth. This aligns training with the inference-time distribution by exposing the model to its own generation errors.
>
> To preserve the model's masked diffusion capability, Stage 3 training interleaves standard masked diffusion (as in Stage 2) with edit-based refinement training. The trade-off between these two components is discussed in Appendix C.2.
>
> > Issue of parallel edit
>
> Edit diffusion remains parallel and factorized at the prediction level. The key difference is the operating regime. Masked diffusion predicts from heavily corrupted inputs, where many positions are uncertain, and independent predictions can lead to inconsistent combinations. In contrast, edit diffusion operates on an already-complete draft, making refinement a low-entropy correction problem under a strong global context.
>
> Although predictions remain factorized, the search space is much more constrained, and inconsistencies are resolved through sparse, localized edits. Empirically, only a small number of edit steps are needed to reach a stable sequence (Section 5.3), indicating that edit diffusion performs correction rather than full regeneration.
>
> > Comparison in Table 1
>
> Due to space constraints, please refer to our response to **Q4 of Reviewer cpgV**, for a discussion of the comparison.
>
> > Evaluation on broader tasks
>
> Due to space constraints, please refer to our response to **W1 of Reviewer 1Eqa**, for a discussion of broader evaluation.

---

> > ### Author Rebuttal · Reviewer_LtYi · 2026-04-03
> >
> > Some of the main weaknesses are not addressed:
> >
> > * Many missing ablations
> > * Comparison in table 1 is still not fair
> > * Code not published, or not clear whether it will be published

---

> > > ### Author Response · Authors · 2026-04-03
> > >
> > > Thank you for your detailed feedback and for the opportunity to further clarify our work. We appreciate the reviewer’s questions, which help us better explain several aspects of the model design and experimental setup.
> > >
> > > > Many missing ablations
> > >
> > > We thank the reviewer for highlighting the importance of additional ablations.
> > >
> > > To further investigate the role of edit-based refinement, we introduce a variant where full edit operations (replacement, deletion, insertion) are allowed from the beginning, without restricting the early stage to mask-only prediction. This corresponds to an edit-only diffusion setting applied throughout the generation process.
> > >
> > > |Budget|Mask Step|Edit Step|HumanEval|MBPP|GSM8k|MATH-500|
> > > |:-:|:-:|:-:|:-:|:-:|:-:|:-:|
> > > |1/1|480|32|**57.9**|**61.6**|84.8|**50.0**|
> > > |1/1|512|0|53.7|60.1|**85.3**|49.6|
> > > |1/1|0|512|31.1|39.9|56.1|29.8|
> > > |1/2|224|32|**51.2**|**54.8**|**82.7**|**50.0**|
> > > |1/2|256|0|50.0|52.6|81.7|48.6|
> > > |1/2|0|256|46.3|53.7|71.5|35.2|
> > > |1/4|96|32|**42.1**|45.0|**76.6**|38.2|
> > > |1/4|128|0|34.8|36.5|71.8|**42.0**|
> > > |1/4|0|128|42.1|**45.8**|74.6|36.4|
> > > |1/8|48|16|25.0|26.7|63.8|**34.4**|
> > > |1/8|64|0|13.4|18.3|53.2|30.8|
> > > |1/8|0|64|**34.1**|**40.5**|**72.5**|29.8|
> > >
> > > From these results, we observe two consistent trends:
> > > - Under low-budget settings (e.g., 1/8), the edit-only variant performs better than mask-only diffusion. This is because edit operations enable larger and more flexible modifications, allowing the model to reach better states more efficiently with limited steps.
> > > - As the number of steps increases, the performance of edit-only diffusion degrades and becomes inferior to the proposed method. We attribute this to a distribution mismatch: during training, edit operations are applied on sequences already refined by mask diffusion, whereas applying edit from the beginning leads to intermediate states that deviate from the training distribution.
> > >
> > > These results support our design choice of using mask diffusion for coarse generation and edit diffusion for refinement, rather than applying edit operations throughout.
> > >
> > > Finally, we would like to emphasize that our ablations are designed to isolate components within the same model and training framework. Together with the ablation of removing edit diffusion (Section 5.3), this reverse ablation (removing mask diffusion) provides a symmetric and controlled analysis of both components. Variants such as training a standalone edit-based generator from scratch (e.g., starting from empty sequences) would correspond to a different modeling formulation and are therefore not directly comparable as controlled ablations.
> > >
> > > > Comparison in table 1 is still not fair
> > >
> > > We agree that cross-method comparisons may not be perfectly fair due to differences in training data.
> > >
> > > For this reason, our primary evaluation focuses on controlled comparisons within the same base model (LLaDA-8B-Base), where Stage-2 and Stage-3 differ only in the proposed edit-based refinement. In particular, Stage-2 and Stage-3 share:
> > > - identical model architecture
> > > - identical training data
> > > - identical inference setup
> > >
> > > This controlled setting isolates the contribution of our method and provides a direct and fair assessment of its effectiveness. The consistent improvements from Stage-2 to Stage-3 across all budgets and tasks demonstrate the benefit of edit-based refinement independent of external factors.
> > >
> > > Following prior work (e.g., Soft Mask [1], EvoToken [2], FlexMDM [3], LRD [4]), we additionally report comparisons with other methods under their respective setups to provide broader context. We will revise the paper to clarify this distinction and explicitly note that such cross-method comparisons should be interpreted with caution.
> > >
> > > [1] Soft-Masked Diffusion Language Models
> > >
> > > [2] Beyond Hard Masks: Progressive Token Evolution for Diffusion Language Models
> > >
> > > [3] Any-Order Flexible Length Masked Diffusion
> > >
> > > [4] Latent Refinement Decoding: Enhancing Diffusion-Based Language Models by Refining Belief States
> > >
> > > > Code not published, or not clear whether it will be published
> > >
> > > We have provided training and inference code in the **Supplementary Material** during the review phase, in accordance with the double-blind policy. We confirm that the full codebase will be publicly released after the review process to ensure reproducibility.

---

### Official Review · Reviewer_cpgV · 2026-03-12

**Soundness:** 3
**Presentation:** 3
**Significance:** 2
**Originality:** 2
**Overall Recommendation:** 4
**Confidence:** 3

**Summary:**

This paper proposes an edit-based refinement framework on top of masked diffusion modeling on text. Instead of relying solely on the diffusion decoding process, the method introduces an additional editing stage that predicts token-level edit operations to refine intermediate generations from conventional decoding. Empirically, the method improves over pure diffusion generation, especially under constrained decoding budgets. The paper is generally well organized and easy to follow, and the ablations provide useful evidence and several interesting observations about the behavior of the proposed refinement procedure.

**Compliance With Llm Reviewing Policy:**

Affirmed.

**Final Justification:**

Please see the rebuttal acknowledgement.

**Key Questions For Authors:**

* [Q1] Since the proposed method appears closer to refinement/correction than to a direct modification of diffusion modeling, why are re-masking or correcting-style approaches not included as the main comparison baselines?

* [Q2] The experiments show that refinement helps, but what evidence supports the specific choice of an edit-based formulation over simpler alternatives such as re-masking or local regeneration?

* [Q3] The refinement training data requires collecting intermediate states from multiple diffusion steps; how expensive is this pipeline in practice, and how well would it scale to larger models and datasets?

* [Q4] What is the reason for using multiple sequential training stages? Can the authors establish a fair baseline under a comparable training procedure?

Each question corresponds to one of the weaknesses discussed above. The authors are welcome to justify both [W]/[Q], or only one of them where appropriate. The [W]s/[Q]s are roughly ordered by their perceived importance from my perspective.

**Limitations:**

Yes.

**Strengths And Weaknesses:**

### Strengths

* Clear empirical gains under controlled decoding budgets.

    The main experimental results consistently show that edit-based refinement improves over pure diffusion generation, especially in low-budget settings where the base diffusion process is more error-prone.

* Well-written and easy to follow.

    The paper is clearly structured, the method is presented in an accessible way, and the main technical ideas are relatively easy to understand.

*  Useful ablations and analysis.

    The ablation studies help support the main empirical claim and also reveal several interesting observations about the refinement design and its practical behavior.

### Weaknesses

* [W1] The most relevant comparisons seem to be missing.

    The proposed method appears conceptually closer to refinement / correction / re-masking style approaches than to methods that directly address the core reverse process of discrete diffusion. While the paper frames the contribution as addressing a fundamental issue of masked discrete diffusion, the actual mechanism is an additional edit-based repair stage on top of generated sequences. For this reason, the most important baselines should include prior refinement-oriented approaches, especially re-masking or correcting methods discussed in the related work. Without these comparisons, it is hard to determine whether the gains come from the specific edit-based formulation, or more generally from adding an extra refinement stage.

* [W2] The paper shows that refinement helps, but does not sufficiently justify why the refinement should take the form of explicit editing.

    The experiments support the usefulness of an additional refinement pass, but they do not convincingly explain why replace/delete/insert operations are the right formulation for the problem. Stronger evidence would be needed to show that edit-based refinement is better motivated than simpler alternatives such as re-masking uncertain tokens, local regeneration, or other generic correction mechanisms.

* [W3] Training data construction may be expensive and difficult to scale.

    The refinement supervision requires collecting intermediate diffusion states across multiple diffusion steps, which may introduce substantial training overhead. This raises concerns about scalability, especially for larger models or more realistic large-scale training settings.

* [W4] The sequential multi-stage training pipeline is somewhat questionable.

    The method relies on several sequential training stages, which makes the pipeline more complex and introduces additional optimization and engineering overhead. This also raises fairness concerns in comparison to the selected baselines, since part of the gain may come from extra stage-specific training rather than the proposed formulation itself.

---

> ### Author Rebuttal · Authors · 2026-03-29
>
> > Q1: Since the proposed method appears closer to refinement/correction than to a direct modification of diffusion modeling, why are re-masking or correcting-style approaches not included as the main comparison baselines?
>
> We thank the reviewer and agree that refinement-based approaches are relevant. We agree that refinement-based approaches are relevant. However, our work focuses on a fixed-budget parallel decoding setting, where many prior refinement methods (e.g., FlexMDM, LRD) rely on adaptive or sequential decoding strategies. As a result, they are not directly comparable under this constraint. In contrast, Soft Mask and EvoToken are designed for this regime and thus serve as more appropriate baselines.
>
> For completeness, we include additional comparisons below (all based on LLaDA; numbers denote relative decoding ratio, where 2.0× uses half the diffusion steps):
>
> |Method|HumanEval|MBPP|GSM8k|MATH-500|
> |:-:|:-:|:-:|:-:|:-:|
> |FlexMDM|-|-|67.0 (1.0×)|-|
> |LRD|48.4 (1.3×)|40.6 (1.5×)|84.5 (2.0×)|39.8 (1.4×)|
> |ME-DLM|57.9 (1.0×)|61.6 (1.0×)|84.8 (1.0×)|50.0 (1.0×)|
> |ME-DLM|51.2 (2.0×)|54.8 (2.0×)|82.7 (2.0×)|50.0 (2.0×)|
>
> These results suggest that even when including refinement-based methods under their own settings, our method remains competitive or better. We also note that some prior works are less directly comparable. For example, Edit-Flow trains a smaller model from scratch, while RemeDi is based on block diffusion and is not evaluated under the same parallel decoding setting.
>
> > Q2: The experiments show that refinement helps, but what evidence supports the specific choice of an edit-based formulation over simpler alternatives such as re-masking or local regeneration?
>
> Our choice of an edit-based formulation is motivated by the specific error patterns observed in parallel masked diffusion. As shown in Fig. 5/6, errors are often structural (e.g., missing tokens or misaligned spans), which inherently require insertion or deletion operations rather than token-level re-prediction.
>
> In such cases, re-masking or local regeneration becomes inefficient or impractical. For example, inserting or deleting a token would require re-masking and regenerating all subsequent positions, effectively rewriting the entire suffix. This is particularly problematic under a fixed diffusion budget, where such cascading updates are difficult to control.
>
> In contrast, our edit-based formulation directly models replacement, insertion, and deletion as localized operations conditioned on the full sequence. This enables targeted corrections without re-generating unaffected regions, making it both more efficient and better aligned with the structural inconsistencies arising from parallel decoding.
>
> > Q3: The refinement training data requires collecting intermediate states from multiple diffusion steps; how expensive is this pipeline in practice, and how well would it scale to larger models and datasets?
>
> We agree that constructing refinement supervision introduces additional training cost, as it requires collecting intermediate diffusion states. However, this overhead is incurred only once during training, and does not affect inference efficiency. In fact, our method enables strong performance under significantly reduced diffusion steps at inference, resulting in lower overall deployment cost.
>
> We view this as a favorable training–inference trade-off. While scaling to larger models would increase the cost of collecting intermediate states, this process is fully parallelizable and follows the same data pipeline as standard diffusion training. Exploring more efficient supervision strategies is an interesting direction for future work.
>
> > Q4: What is the reason for using multiple sequential training stages? Can the authors establish a fair baseline under a comparable training procedure?
>
> The multi-stage training design is used to progressively introduce the capabilities required for edit-based refinement. Importantly, we ensure a controlled comparison between Stage-2 (mask diffusion only) and Stage-3 (mask + edit diffusion): both are trained on the same data, model, and pipeline, differing only in the addition of edit-based refinement. Therefore, Stage-2 serves as a direct and fair baseline for isolating the contribution of our method.
>
> For other approaches, reproducing them under the same multi-stage pipeline would require re-training large-scale models with different objectives, which is computationally expensive and beyond the scope of this work. Instead, we ensure fairness by comparing methods under matched or normalized decoding budgets and focusing on approaches designed for similar parallel decoding settings.

---

> > ### Author Rebuttal · Reviewer_cpgV · 2026-04-03
> >
> > I think the authors have addressed Q1/Q2 to some extend. For Q3, the authors also agree that the training introduces additional cost. Including an ablation regarding this part would help strengthen the completeness of the paper. Q4 seems not easy to address in a short rebuttal. Overall, I would like to maintain my (favorable) score.

---

> > > ### Author Response · Authors · 2026-04-07
> > >
> > > > Q3: ablation experiments on training cost
> > >
> > > Thank you for this helpful suggestion. We agree that the most relevant question here is whether the extra cost of obtaining model-generated intermediate states is necessary for effective edit-refinement training. To investigate this, we add two additional variants:
> > > - **Mask-only states**. We always set the edit rollout depth to n = 0, i.e., the model is trained only on the intermediate state obtained from mask diffusion, without performing edit diffusion rollout during training. This removes the additional cost of collecting deeper model-generated edit states while preserving the mask-diffusion initialization. Since the average number of steps for the mask is much greater than that for the edit, this variant offers only a small reduction in training cost, decreasing training time by approximately 10%.
> > > - **Rule-based noise**. Instead of obtaining intermediate states from model rollout, we construct noisy inputs by randomly inserting, deleting, and replacing tokens, and train the model to recover the target sequence from these corrupted inputs. This variant removes the training-time inference cost of intermediate-state construction entirely.
> > >
> > > |Budget|Method|HumanEval|MBPP|GSM8k|MATH-500|
> > > |:-:|:-:|:-:|:-:|:-:|:-:|
> > > |1/1|ME-DLM Stage 2|56.1|54.8|80.9|45.8|
> > > |1/1|ME-DLM Stage 3|**57.9**|**61.6**|**84.8**|**50.0**|
> > > |1/1|w/ mask-only states|52.4|57.9|84.2|49.4|
> > > |1/1|w/ rule-based noise|54.3|56.3|78.8|44.8|
> > > |1/2|ME-DLM Stage 2|45.7|52.4|76.3|43.0|
> > > |1/2|ME-DLM Stage 3|**51.2**|**54.8**|**82.7**|**50.0**|
> > > |1/2|w/ mask-only states|48.8|48.7|81.1|49.0|
> > > |1/2|w/ rule-based noise|47.0|53.7|72.6|40.8|
> > > |1/4|ME-DLM Stage 2|36.6|34.7|58.9|32.0|
> > > |1/4|ME-DLM Stage 3|**42.1**|**45.0**|**76.6**|38.2|
> > > |1/4|w/ mask-only states|38.4|33.9|74.0|**40.4**|
> > > |1/4|w/ rule-based noise|39.0|42.9|55.6|32.6|
> > > |1/8|ME-DLM Stage 2|13.4|22.0|30.2|17.8|
> > > |1/8|ME-DLM Stage 3|**25.0**|**26.7**|**63.8**|**34.4**|
> > > |1/8|w/ mask-only states|**25.0**|25.9|61.4|31.0|
> > > |1/8|w/ rule-based noise|21.3|21.2|29.6|18.0|
> > >
> > > As shown in the table, we have several observations:
> > > - The rule-based noise variant performs much worse on the math benchmarks. This is mainly because the perturbation rules are random, and therefore often fail to modify the final answer span, which is exactly where many math errors occur. As a result, the resulting noisy sequences do not match the error distribution at test time, leading to a clear train–test mismatch.
> > > - For mask-only states, the training cost is reduced, but the performance also drops. This drop is especially more visible on code tasks, in some cases even below Stage-2. This is because code generation is highly sensitive to token-level errors: even a single wrong token can make the entire output incorrect. Therefore, the mismatch between training states and test-time intermediate states has a larger impact on code tasks.
> > >
> > > Overall, these results suggest that the key issue is not simply reducing the cost of constructing intermediate states, but reducing the cost of obtaining intermediate states that better match the test-time distribution. We view this as an important direction for future work.

---

### Official Review · Reviewer_eUM7 · 2026-03-13

**Soundness:** 3
**Presentation:** 3
**Significance:** 3
**Originality:** 3
**Overall Recommendation:** 5
**Confidence:** 3

**Summary:**

This paper proposes ME-DLM, an edit-based refinement framework for diffusion language models. It augments diffusion generation with lightweight post-editing steps: after producing an initial complete response, the model refines it through edit operations such as insertion, deletion, and replacement. The experiments demonstrate that the proposed method improves the quality and robustness of generation: the method achieves consistent gains of 11.6% on HumanEval and 33.6% on GSM8K over LLaDA while using one-eighth of the total diffusion steps.

**Compliance With Llm Reviewing Policy:**

Affirmed.

**Final Justification:**

I don't see the major weakness of this paper. The rebuttal does not change my positive evaluation of this paper.

**Key Questions For Authors:**

Please explain how the ideas proposed in this work could be applied to practical diffusion language models (DLMs) such as Block Diffusion.

**Limitations:**

While the paper discusses the social impact, it does not address the limitations of the proposed method. If there are weaknesses or challenges relative to existing approaches, the authors should include a discussion of these limitations.

**Strengths And Weaknesses:**

### Soundness

The proposed approach, which predicts edits, is a reasonable method for improving sequence-level consistency in Diffusion Language Models. The superiority of the proposed method is demonstrated through comparisons with prior work and with LLaDA. In addition, analyses of the experimental results are reported. Overall, the study appears to be sound.

### Presentation

The paper is clear and easy to read. The background of Diffusion Language Models is also well explained (although it may be unnecessary for readers already familiar with DLMs). The motivation for the proposed approach is described with concrete examples. The proposed method is appropriately explained using equations and figures.

### Significance

Diffusion Language Models have recently attracted significant attention as a research area, and further developments in this field may improve the efficiency of LLM inference.

### Originality

The idea of having a Diffusion Language Model predict edit operations is not particularly surprising, but to the best of my knowledge, there has been little prior work exploring this direction. That said, it is unclear whether predicting edit operations should be described as diffusion or denoising.

---

> ### Author Rebuttal · Authors · 2026-03-29
>
> > Q1: Please explain how the ideas proposed in this work could be applied to practical diffusion language models (DLMs) such as Block Diffusion.
>
> We thank the reviewer for this insightful question. Our method can be naturally extended to practical diffusion language models such as block diffusion, and we agree that a key consideration is how to integrate refinement into the block-wise generation paradigm.
>
> A straightforward yet effective approach is to introduce a two-stage refinement mechanism that operates at both the block level and the full-sequence level:
>
> Intra-block refinement. Within each block, we first perform standard masked diffusion (i.e., parallel unmasking) to obtain a coarse local sequence. We then apply edit-based diffusion within the block to correct local inconsistencies introduced by parallel decoding. This preserves the efficiency and parallelism of block diffusion while improving intra-block coherence.
>
> Inter-block (global) refinement. After all blocks are generated, we perform an additional round of edit-based diffusion over the entire concatenated sequence. This stage is particularly important because block-wise generation can introduce boundary inconsistencies across blocks (e.g., mismatched dependencies or incomplete structures). Since our edit-based refinement is conditioned on the full sequence, it is well-suited to resolving such cross-block inconsistencies through minimal global edits.
>
> Importantly, this design aligns closely with our formulation: the initial masked diffusion stage produces a coarse but potentially inconsistent sequence, while the edit-based diffusion stage performs globally conditioned corrections with minimal local edits. Notably, our method does not require changes to the underlying diffusion objective or architecture, and can be incorporated as a modular refinement component on top of existing block diffusion systems.
>
> Overall, our method is particularly well-suited to block diffusion, as it directly targets the key limitation of parallel generation—namely, the mismatch between token-level predictions and sequence-level consistency—at both local (block) and global (sequence) scales.

---

> > ### Author Rebuttal · Reviewer_eUM7 · 2026-04-04
> >
> > Thank you for answering my question about the generality of the proposed method. I don't have any further questions.

---

### Decision · Program_Chairs · 2026-04-30

**Decision:**

Accept (regular)

**Comment:**

This paper proposes an edit-based refinement method for masked diffusion LMs, aimed at improving sequence level consistency. After diffusion decodding, a lightweight edit-based refinement stage performs insertion, deletion, and replacement operations conditioned on the full sequence. Across reviewers, there is agreement that the paper is technically sound and empirically effective.

Reviewers raise a concern that the paper does not fully establish why the specific edit-based formulation is preferable to simpler alternatives such as re-masking or local regeneration. Relatedly, comparisons to prior refinement-style baselines are limited, making it difficult to isolate the contribution of the specific design choices. Other concerns relate to training complexity as well as limited evaluation breadth beyond reasoning and code tasks. The authors have provided acceptable responses to most of these concerns.

I therefore recommend acceptanc and encourage the authors to strengthen the discussion of related refinement approaches, training cost, and evaluation breadth in the final version.